# PROGRESSIVE LEARNING AND DISENTANGLEMENT OF HIERARCHICAL REPRESENTATIONS

**Zhiyuan Li, Jaideep Vitthal Murkute, Prashnna Kumar Gyawali & Linwei Wang**
Golisano College of Computing and Information Sciences
Rochester Institute of Technology
Rochester, NY 14623, USA
`{zl7904,jvm6526,pkg2182,Linwei.Wang}@rit.edu`

## ABSTRACT

Learning rich representation from data is an important task for deep generative models such as variational auto-encoder (VAE). However, by extracting high-level abstractions in the bottom-up inference process, the goal of preserving all factors of variations for top-down generation is compromised. Motivated by the concept of "starting small", we present a strategy to progressively learn independent hierarchical representations from high- to low-levels of abstractions. The model starts with learning the most abstract representation, and then progressively grow the network architecture to introduce new representations at different levels of abstraction. We quantitatively demonstrate the ability of the presented model to improve disentanglement in comparison to existing works on two benchmark data sets using three disentanglement metrics, including a new metric we proposed to complement the previously-presented metric of mutual information gap. We further present both qualitative and quantitative evidence on how the progression of learning improves disentangling of hierarchical representations. By drawing on the respective advantage of hierarchical representation learning and progressive learning, this is to our knowledge the first attempt to improve disentanglement by progressively growing the capacity of VAE to learn hierarchical representations[1].

## 1 INTRODUCTION

Variational auto-encoder (VAE), a popular deep generative model (DGM), has shown great promise in learning interpretable and semantically meaningful representations of data (Higgins et al. (2017); Chen et al. (2018); Kim & Mnih (2018); Gyawali et al. (2019)). However, VAE has not been able to fully utilize the depth of neural networks like its supervised counterparts, for which a fundamental cause lies in the inherent conflict between the bottom-up inference and top-down generation process (Zhao et al. (2017); Li et al. (2016)): while the bottom-up abstraction is able to extract high-level representations helpful for discriminative tasks, the goal of generation requires the preservation of all generative factors that are likely at different abstraction levels. This issue was addressed in recent works by allowing VAEs to generate from details added at different depths of the network, using either memory modules between top-down generation layers (Li et al. (2016)), or hierarchical latent representations extracted at different depths via a variational ladder autoencoder (VLAE, Zhao et al. (2017)).

However, it is difficult to learn to extract and disentangle all generative factors at once, especially at different abstraction levels. Inspired by human cognition system, Elman (1993) suggested the importance of "starting small" in two aspects of the learning process of neural networks: *incremental input* in which a network is trained with data and tasks of increasing complexity, and *incremental memory* in which the network capacity undergoes developmental changes given fixed external data and tasks — both pointing to an incremental learning strategy for simplifying a complex final task. Indeed, the former concept of *incremental input* has underpinned the success of curriculum learning (Bengio et al. (2015)). In the context of DGMs, various stacked versions of generative adversarial networks (GANs) have been proposed to decompose the final task of high-resolution image generation into progressive sub-tasks of generating small to large images (Denton et al. (2015); Zhang

---

[1]Source code available at `https://github.com/Zhiyuan1991/proVLAE`.

et al. (2018)). The latter aspect of "starting small" with incremental growth of network capacity is less explored, although recent works have demonstrated the advantage of progressively growing the depth of GANs for generating high-resolution images (Karras et al. (2018); Wang et al. (2018)). These works, so far, have focused on progressive learning as a strategy to improve image generation.

We are motivated to investigate the possibility to use progressive learning strategies to improve learning and disentangling of hierarchical representations. At a high level, the idea of progressively or sequentially learning latent representations has been previously considered in VAE. In Gregor et al. (2015), the network learned to sequentially refine generated images through recurrent networks. In Lezama (2019), a teacher-student training strategy was used to progressively increase the number of latent dimensions in VAE to improve the generation of images while preserving the disentangling ability of the teacher model. However, these works primarily focus on progressively growing the capacity of VAE to generate, rather than to extract and disentangle hierarchical representations.

In comparison, in this work, we focus on 1) progressively growing the capacity of the network to extract hierarchical representations, and 2) these hierarchical representations are extracted and used in generation from different abstraction levels. We present a simple progressive training strategy that grows the hierarchical latent representations from different depths of the inference and generation model, learning from high- to low-levels of abstractions as the capacity of the model architecture grows. Because it can be viewed as a progressive strategy to train the VLAE presented in Zhao et al. (2017), we term the presented model *pro*-VLAE. We quantitatively demonstrate the ability of *pro*-VLAE to improve disentanglement on two benchmark data sets using three disentanglement metrics, including a new metric we proposed to complement the metric of mutual information gap (MIG) previously presented in Chen et al. (2018). These quantitative studies include comprehensive comparisons to $\beta$-VAE (Higgins et al. (2017)), VLAE (Zhao et al. (2017)), and the teacher-student strategy as presented in (Lezama (2019)) at different values of the hyperparameter $\beta$. We further present both qualitative and quantitative evidence that *pro*-VLAE is able to first learn the most abstract representations and then progressively disentangle existing factors or learn new factors at lower levels of abstraction, improving disentangling of hierarhical representations in the process.

## 2 RELATED WORKS

A hierarchy of feature maps can be naturally formed in stacked discriminative models (Zeiler & Fergus (2014)). Similarly, in DGM, many works have proposed stacked-VAEs as a common way to learn a hierarchy of latent variables and thereby improve image generation (Sønderby et al. (2016); Bachman (2016); Kingma et al. (2016)). However, this stacked hierarchy is not only difficult to train as the depths increases (Sønderby et al. (2016); Bachman (2016)), but also has an unclear benefit for learning either hierarchical or disentangled representations: as shown in Zhao et al. (2017), when fully optimized, it is equivalent to a model with a single layer of latent variables. Alternatively, instead of a hierarchy of latent variables, independent hierarchical representations at different abstraction levels can be extracted and used in generation from different depths of the network (Rezende et al. (2014); Zhao et al. (2017)). A similar idea was presented in Li et al. (2016) to generate lost details from memory and attention modules at different depths of the top-down generation process. The presented work aligns with existing works (Rezende et al. (2014); Zhao et al. (2017)) in learning independent hierarchical representation from different levels of abstraction, and we look to facilitate this learning by progressively learning the representations from high- to low-levels.

Progressive learning has been successful for high-quality image generation, mostly in the setting of GANs. Following the seminar work of Elman (1993), these progressive strategies can be loosely grouped into two categories. Mostly, in line with *incremental input*, several works have proposed to divide the final task of image generation into progressive tasks of generating low-resolution to high-resolution images with multi-scale supervision (Denton et al. (2015); Zhang et al. (2018)). Alternatively, in line with *incremental memory*, a small number of works have demonstrated the ability to simply grow the architecture of GANs from a shallow network with limited capacity for generating low-resolution images, to a deep network capable of generating super-resolution images (Karras et al. (2018); Wang et al. (2018)). This approach was also shown to be time-efficient since the early-stage small networks require less time to converge comparing to training a full network from the beginning. This latter group of works provided compelling evidence for the benefit of progressively growing the capacity of a network to generate images, although its extension for growing the capacity of a network to learn hierarchical representations has not been explored.

Limited work has considered incremental learning of representations in VAE. In Gregor et al. (2015), recurrent networks with attention mechanisms were used to sequentially refines the details in generated images. It however focused on the generation performance of VAE without considering the learned representations. In Lezama (2019), a teacher-student strategy was used to progressively grow the dimension of the latent representations in VAE. Its fundamental motivation was that, given a teacher model that has learned to effectively disentangle major factors of variations, progressively learning additional nuisance variables will improve generation without compromising the disentangling ability of the teacher – the latter accomplished via a newly-proposed Jacobian supervision. The capacity of this model to grow, thus, is by design limited to the extraction of nuisance variables. In comparison, we are interested in a more significant growth of the VAE capacity to progressively learn and improve disentangling of important factors of variations which, as we will later demonstrate, is not what the model in Lezama (2019) is intended for. In addition, neither of these works considered learning different levels of abstractions at different depths of the network, and the presented *pro*-VLAE provides a simpler training strategy to achieve progressive representation learning.

Learning disentangled representation is a primary motivation of our work, and an important topic in VAE. Existing works mainly tackle this by promoting the independence among the learned latent factors in VAE (Higgins et al. (2017); Kim & Mnih (2018); Chen et al. (2018)). The presented progressive learning strategy provides a novel approach to improve disentangling that is different to these existing methods and a possibility to augment them in the future.

## 3 METHODS

### 3.1 MODEL: VAE WITH HIERARCHICAL REPRESENTATIONS

We assume a generative model $p(\boldsymbol{x}, \boldsymbol{z}) = p(\boldsymbol{x}|\boldsymbol{z})p(\boldsymbol{z})$ for observed $\boldsymbol{x}$ and its latent variable $\boldsymbol{z}$. To learn hierarchical representations of $\boldsymbol{x}$, we decompose $\boldsymbol{z}$ into $\{\boldsymbol{z}_1, \boldsymbol{z}_2, ..., \boldsymbol{z}_L\}$ with $\boldsymbol{z}_l(l = 1, 2, 3, ..., L)$ from different abstraction levels that are loosely guided by the depth of neural network as in Zhao et al. (2017). We define the hierarchical generative model $p_\theta$ as:

$$p(\boldsymbol{x}, \boldsymbol{z}) = p(\boldsymbol{x}|\boldsymbol{z}_1, \boldsymbol{z}_2, ..., \boldsymbol{z}_L) \prod_{l=1}^{L} p(\boldsymbol{z}_l). \tag{1}$$

Note that there is no hierarchical dependence among the latent variables as in common hierarchical latent variable models. Rather, similar to that in Rezende et al. (2014) and Zhao et al. (2017), $\boldsymbol{z}_l$'s are independent and each represents generative factors at an abstraction level not captured in other levels. We then define an inference model $q_\phi$ to approximate the posterior as:

$$q(\boldsymbol{z}_1, \boldsymbol{z}_2, ..., \boldsymbol{z}_L|\boldsymbol{x}) = \prod_{l=1}^{L} q(\boldsymbol{z}_l|\boldsymbol{h}_l(\boldsymbol{x})), \tag{2}$$

where $\boldsymbol{h}_l(\boldsymbol{x})$ represents a particular level of bottom-up abstraction of $\boldsymbol{x}$. We parameterize $p_\theta$ and $q_\phi$ with an encoding-decoding structure and, as in Zhao et al. (2017), we approximate the abstraction level with the network depth. The full model is illustrated in Fig. 1(c), with a final goal to maximize a modified evidence lower bound (ELBO) of the marginal likelihood of data $\boldsymbol{x}$:

$$\log p(\boldsymbol{x}) \geq \mathcal{L} = \mathbb{E}_{q(\boldsymbol{z}|\boldsymbol{x})}[\log p(\boldsymbol{x}|\boldsymbol{z})] - \beta KL(q(\boldsymbol{z}|\boldsymbol{x})||p(\boldsymbol{z})), \tag{3}$$

where $KL$ denotes the Kullback-Leibler divergence, prior $p(\boldsymbol{z})$ is set to isotropic Gaussian $\mathcal{N}(0, \boldsymbol{I})$ according to standard practice, and $\beta$ is a hyperparameter introduced in Higgins et al. (2017) to promote disentangling, defaulting to the standrd ELBO objective when $\beta = 1$.

### 3.2 PROGRESSIVE LEARNING OF HIERARCHICAL REPRESENTATION

We present a progressive learning strategy, as illustrated in Fig. 1, to achieve the final goal in equation (3) by learning the latent variables $\boldsymbol{z}_l$ progressively from the highest ($l = L$) to the lowest $l = 1$) level of abstractions. We start by learning the most abstraction representations at layer $L$ as show in Fig. 1(a). In this case, our model degenerates to a vanilla VAE with latent variables $\boldsymbol{z}_L$ at the deepest layer. We keep the dimension of $\boldsymbol{z}_L$ small to start small in terms of the capacity to learn latent representations, where we define the inference model at progressive step $s = 0$ as:

$$\boldsymbol{z}_L \sim \mathcal{N}(\mu_L(\boldsymbol{h}_L), \sigma_L(\boldsymbol{h}_L)), \ \boldsymbol{h}_l = f_l^e(\boldsymbol{h}_{l-1}), \text{ for } l = 1, 2, ..., L, \text{ and } \boldsymbol{h}_0 \equiv \boldsymbol{x}, \tag{4}$$

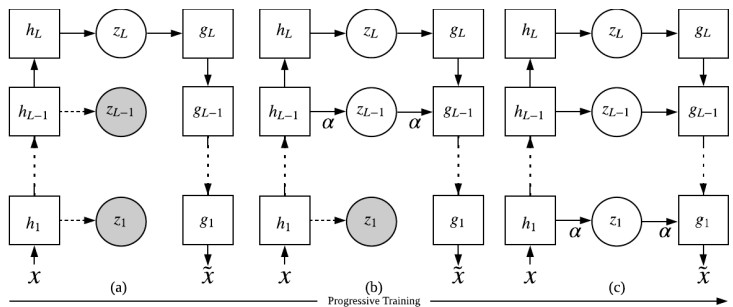

Figure 1: Progressive learning of hierarchical representations. White blocks and solid lines are VAE models at the current progression. $\alpha$ is a fade-in coefficient for blending in the new network component. Gray circles and dash line represents (optional) constraining of the future latent variables.

and the generative model as:

$$\boldsymbol{g}_L = f_L^d(\boldsymbol{z}_L), \; \boldsymbol{g}_l = f_l^d(\boldsymbol{g}_{l+1}), \; \boldsymbol{x} = D(\boldsymbol{x}; f_0^d(\boldsymbol{g}_0)), \tag{5}$$

where $f_l^e$, $\mu_L$, and $\sigma_L$ are parts of the encoder architecture, $f_l^d$ are parts of the decoder architecture, and $D$ is the distribution of $\boldsymbol{x}$ parametrized by $f_0^d(\boldsymbol{g}_0)$, which can be either Bernoulli or Gaussian depending on the data. Next, as shown in Fig. 1, we progressively grow the model to learn $z_{L-1}, ..., z_2, z_1$ from high to low abstraction levels. At each progressive step $s = 1, 2, ..., L - 1$, we move down one abstraction level, and grow the inference model by introducing new latent code:

$$\boldsymbol{z}_l \sim \mathcal{N}(\mu_l(\boldsymbol{h}_l), \sigma_l(\boldsymbol{h}_l)), \; l = L - s. \tag{6}$$

Simultaneously, we grow the decoder such that it can generate with the new latent code as:

$$\boldsymbol{g}_l = f_l^d([m_l(\boldsymbol{z}_l); \boldsymbol{g}_{l+1}]), \; l = L - s, \tag{7}$$

where $m_l$ includes transposed convolution layers outputting a feature map in the same shape as $\boldsymbol{g}_{l+1}$, and $[\cdot ; \cdot]$ denotes a concatenation operation. The training objective at progressive step $s$ is then:

$$\mathcal{L}_{pro} = \mathbb{E}_{q(\boldsymbol{z}_L, \boldsymbol{z}_{L-1}, ..., \boldsymbol{z}_{L-s}|\boldsymbol{x})}[\log p(\boldsymbol{x}|\boldsymbol{z}_L, \boldsymbol{z}_{L-1}, ..., \boldsymbol{z}_{L-s})] - \beta \sum_{L-s}^{L} KL(q(\boldsymbol{z}_l|\boldsymbol{x})||p(\boldsymbol{z}_l)), \tag{8}$$

By replacing the full objective in equation (3) with a sequence of the objectives in equation (8) as the training progresses, we incrementally learn to *extract* and *generate with* hierarchical latent representations $z_l$'s from high to low levels of abstractions. Once trained, the full model as shown in Fig. 1(c) will be used for inference and generation, and progressive processes are no loner needed.

### 3.3 IMPLEMENTATION STRATEGIES

Two important strategies are utilized to implement the proposed progressive representation learning. First, directly adding new components to a trained network often introduce a sudden shock to the gradient: in VAEs, this often leads to the explosion of the variance in the latent distributions. To avoid this shock, we adopt the popular method of "fade-in" (Karras et al. (2018)) to smoothly blend the new and existing network components. In specific, we introduce a "fade-in" coefficient $\alpha$ to equations (6) and (7) when growing new components in the encoder and the decoder:

$$\boldsymbol{z}_l \sim \mathcal{N}(\mu_l(\alpha \boldsymbol{h}_l), \; \sigma_l(\alpha \boldsymbol{h}_l)), \boldsymbol{g}_l = f_l^d([\alpha m_l(\boldsymbol{z}_l); \boldsymbol{g}_{l+1}]), \tag{9}$$

where $\alpha$ increases from 0 to 1 within a certain number of iterations (5000 in our experiments) since the addition of the new network components $\mu_l, \sigma_l$, and $m_l$.

Second, we further stabilize the training by weakly constraining the distribution of $z_l$'s before they are added to the network. This can be achieved by a applying a KL penalty, modulated by a small coefficient $\gamma$, to all latent variables that have not been used in the generation at progressive step $s$:

$$\mathcal{L}_{pre-trained} = \gamma \sum_{l=1}^{L-s-1} \big[ - KL(q(\boldsymbol{z}_l|\boldsymbol{x})||p(\boldsymbol{z}_l)) \big], \tag{10}$$

where $\gamma$ is set to 0.5 in our experiments. The final training objective at step $s$ then becomes:

$$\mathcal{L} = \mathcal{L}_{pro} + \mathcal{L}_{pre-trained} \tag{11}$$

Note that the latent variables at the hierarchy lower than $L - s$ are neither meaningfully inferred nor used in generation at progressive step $s$, and $\mathcal{L}_{pre-trained}$ merely intends to regularize the distribution of these latent variables before they are added to the network. In the experiments below, we use both "fade-in" and $\mathcal{L}_{pre-trained}$ when implementing the progressive training strategy.

## 3.4 DISENTANGLEMENT METRIC

Various quantitative metrics for measuring disentanglement have been proposed (Higgins et al. (2017); Kim & Mnih (2018); Chen et al. (2018)). For instance, the recently proposed MIG metrics (Chen et al. (2018)) measures the gap of mutual information between the top two latent dimensions that have the highest mutual information with a given generative factor. A low MIG score, therefore, suggests an undesired outcome that the same factor is split into multiple dimensions. However, if different generative factors are entangled into the same latent dimension, the MIG score will not be affected.

Therefore, we propose a new disentanglement metric to supplement MIG by recognizing the entanglement of multiple generative factors into the same latent dimension. We define MIG-sup as:

$$\frac{1}{J} \sum_{1}^{J} \left( I_{norm}(\boldsymbol{z}_j; v_{k^{(j)}}) - \max_{k \neq k^{(j)}} I_{norm}(\boldsymbol{z}_j; v_k) \right), \tag{12}$$

where $\boldsymbol{z}$ is the latent variables and $v$ is the ground truth factors, $k^{(j)} = \operatorname{argmax}_k I_{norm}(\boldsymbol{z}_j; v_k)$, $J$ is the number of meaningful latent dimensions, and $I_{norm}(\boldsymbol{z}_j; v_k)$ is normalized mutual information $I(\boldsymbol{z}_j; v_k)/H(v_k)$. Considering MIG and MIG-sup together will provide a more complete measure of disentanglement, accounting for both the splitting of one factor into multiple dimensions and the encoding of multiple factors into the same dimension. In an ideal disentanglement, both MIG and MIG-sup should be 1, recognizing a one-to-one relationship between a generative factor and a latent dimension. This would have a similar effect to the metric that was proposed in Eastwood & Williams (2018), although MIG-based metrics do not rely on training extra classifiers or regressors and are unbiased for hyperparameter settings. The factor metric (Kim & Mnih (2018)) also has similar properties with MIG-sup, although MIG-sup is stricter on penalizing any amount of other minor factors in the same dimension.

## 4 EXPERIMENT

We tested the presented *pro*-VLAE on four benchmark data sets: dSprites (Matthey et al. (2017)), 3DShapes (Burgess & Kim (2018)), MNIST (LeCun et al. (1998)), and CelebA (Liu et al. (2015)), where the first two include ground-truth generative factors that allow us to carry out comprehensive quantitative comparisons of disentangling metrics with existing models. In the following, we first quantitatively compare the disentangling ability of *pro*-VLAE in comparison to three existing models using three disentanglement metrics. We then analyze *pro*-VLAE from the aspects of how it learns progressively, its ability to disentangle, and its ability to learn abstractions at different levels.

**Comparisons in quantitative disentanglement metrics:** For quantitative comparisons, we considered the factor metric in Kim & Mnih (2018), the MIG in Chen et al. (2018), and the MIG-sup presented in this work. We compared *pro*-VLAE (changing $\beta$) with beta-VAE (Higgins et al. (2017)), VLAE (Zhao et al. (2017)) as a hierarchical baseline without progressive training, and the teacher-student model (Lezama (2019)) as the most related progressive VAE without hierarchical representations. All models were considered at different values of $\beta$ except the teacher-student model: the comparison of $\beta$-VAE, VLAE, and the presented *pro*-VLAE thus also provides an ablation study on the effect of learning hierarchical representations and doing so in a progressive manner.

For fair comparisons, we strictly required all models to have the same number of latent variables and the same number of training iterations. For instance, if a hierarchical model has three layers that each has three latent dimensions, a non-hierarchical model will have nine latent dimensions; if a progressive method has three progressive steps with 15 epochs of training each, a non-progressive method will be trained for 45 epochs. Three to five experiments were conducted for each model at each $\beta$ value, and the average of the top three is used for reporting the quantitative results in Fig. 2.

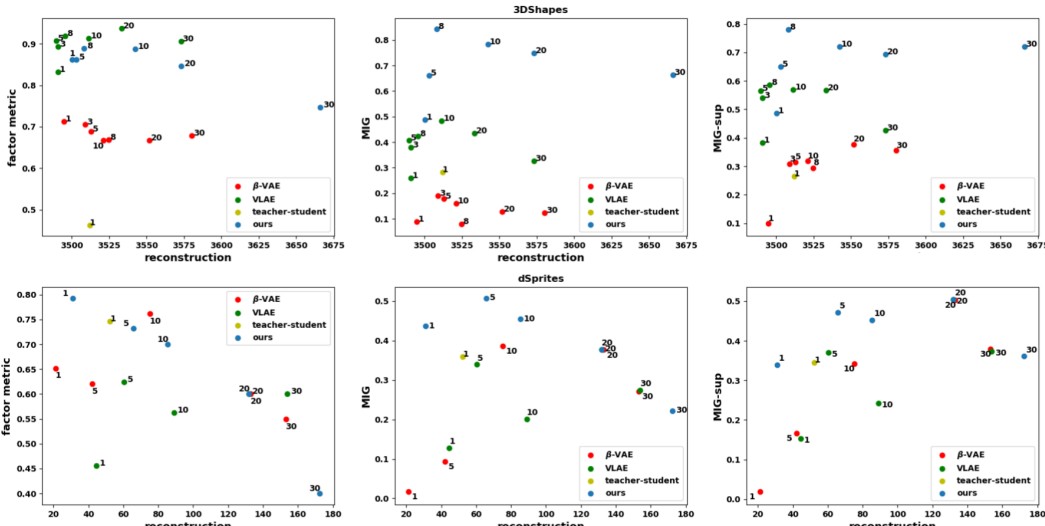

Figure 2: Quantitative comparison of disentanglement metrics. Each point is annotated by the $\beta$ value and averaged over top three best random seeds for the given $\beta$ on the give model. **Left to right:** reconstruction errors *vs.* disentanglement metrics of factor, MIG, and MIG-sup, a higher value indicating a better disentanglement in each metric.

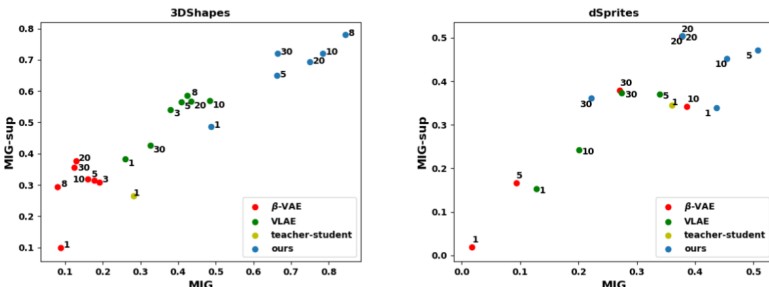

Figure 3: MIG *vs.* MIG-sup following a similar presentation in Fig. 2. A better disentanglement should have higher MIG and higher MIG-sup, locating at the top-right quadrant of the plot.

As shown, for MIG and MIG-sup, VLAE generally outperformed $\beta$-VAE at most $\beta$ values, while *pro*-VLAE showed a clear margin of improvement over both methods. With the factor metric, *pro*-VLAE was still among the top performers, although with a smaller margin and a larger overlap with VLAE on 3DShapes, and with $\beta$-VAE ($\beta = 10$) on dSprites. The teacher-student strategy with Jacobian supervision in general had a low to moderate disentangling score, especially on 3DShapes. This is consistent with the original motivation of the method for progressively learning nuisance variables after the teacher learns to disentangle effectively, rather than progressively disentangling hierarchical factors of variations as intended by *pro*-VLAE. Note that *pro*-VLAE in general performed better with a smaller value of $\beta$ ($\beta < 20$), suggesting that progressive learning already had an effect of promoting disentangling and a high value of $\beta$ may over-promote disentangling at the expense of reconstruction quality.

Fig. 3 shows MIG *vs.* MIG-sup scores among the tested models. As shown, results from *pro*-VLAE were well separated from the other three models at the right top quadrant of the plots, obtaining simultaneously high MIG and MIG-sup scores as a clear evidence for improved disentangling ability.

Fig. 4 provides images generated by traversing each latent dimension using the best *pro*-VLAE ($\beta = 8$), the best VLAE ($\beta = 10$), and the teacher-student model on 3DShapes data. As shown, *pro*-VLAE learned to disentangle the object, wall, and floor color in the deepest layer; the following hierarchy of representations then disentangled objective scale, orientation, and shape, while the lowest-level of abstractions ran out of meaningful generative factors to learn. In comparison, the VLAE distributed six generative factors over the nine latent dimensions, where color was split across

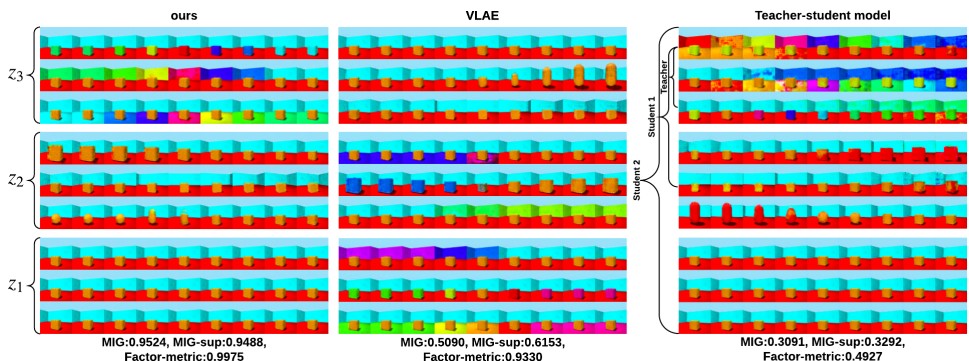

Figure 4: Traversing each latent dimension in *pro*-VLAE ($\beta = 8$), VLAE ($\beta = 10$), and teacher-student model. The hierarchy of the latent variables is noted by brackets on the side.

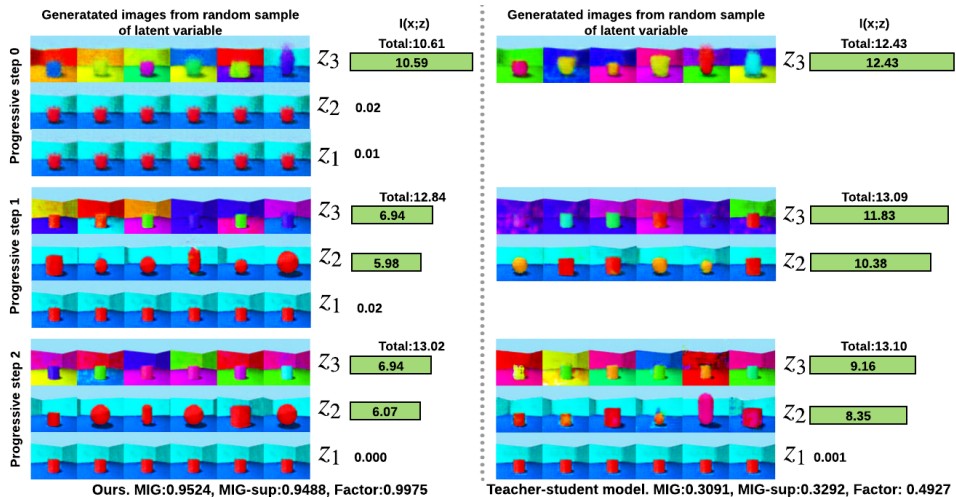

Figure 5: Progressive learning of hierarchical representations. At each progression and for each $z_l$, the row of images are generated by randomly sampling from its prior distributions while fixing the other latent variables (this is NOT traversing). The green bar at each row tracks the mutual information $I(\boldsymbol{x}; \boldsymbol{z}_l)$, while the total mutual information $I(\boldsymbol{x}; \boldsymbol{z})$ is labeled on top.

the hierarchy and sometimes entangled with the object scale (in $\boldsymbol{z}_2$). The teacher-student model was much less disentangled, which we will delve into further in the following section.

**Information flow during progressive learning:** To further understand what happened during progressive learning, we use mutual information $I(\boldsymbol{x}, \boldsymbol{z}_l)$ as a surrogate to track the amount of information learned in each hierarchy of latent variables $\boldsymbol{z}_l$ during the progressive learning. We adopted the approach in Chen et al. (2018) to empirically estimate the mutual information by stratified sampling.

Fig. 5 shows an example from 3DShapes. At progressive step 0, *pro*-VAE was only learning the deepest latent variables in $\boldsymbol{z}_3$, discovering most of the generative factors including color, objective shape, and orientation entangled within $\boldsymbol{z}_3$. At progressive step 1, interestingly, the model was able to "drag" out shape and rotation factors from $\boldsymbol{z}_3$ and disentangle them into $\boldsymbol{z}_2$ along with a new scale factor. Thus $I(\boldsymbol{x}; \boldsymbol{z}3)$ decreased from 10.59 to 6.94 while $I(\boldsymbol{x}; \boldsymbol{z}2)$ increased from 0.02 to 5.98 in this progression, while the total mutual information $I(\boldsymbol{x}; \boldsymbol{z})$ increased from 10.61 to 12.84, suggesting the overall learning of more detailed information. Since 3DShapes only has 6 factors, the lowest-level representation $\boldsymbol{z}_1$ had nothing to learn in progressive step 2, and the allocation of mutual information remained nearly unchanged. Note that the sum of $I(\boldsymbol{x}, \boldsymbol{z}_l)$'s does not equal to $I(\boldsymbol{x}, \boldsymbol{z})$ and $I_{over} = \sum_1^L I(\boldsymbol{x}, \boldsymbol{z}_l) - I(\boldsymbol{x}, \boldsymbol{z})$ suggests the amount of information that is entangled.

In comparison, the teacher-student model was less effective in progressively dragging entangled representations to newly added latent dimensions, as suggested by the slowing changing of $I(\boldsymbol{x}, \boldsymbol{z}_l)$'s

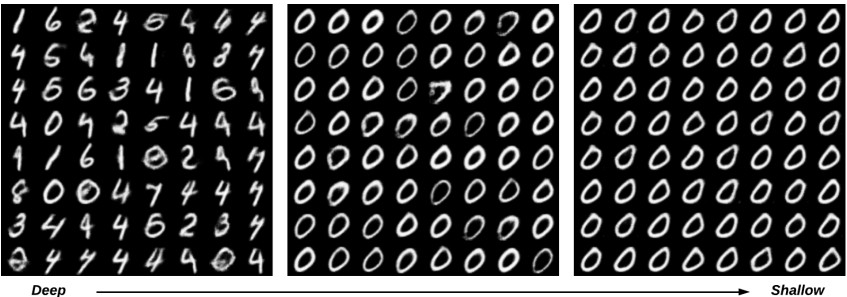

Figure 6: Visualization of hierarchical features learnt for MNIST data. Each sub-figure is generated by randomly sampling from the prior distribution of $z_l$ at one abstraction level while fixing the others. The original latent code is inferred from a image with digit "0". **From left to right:** $z_3$ encodes the highest abstraction: digit identity; $z_2$ encodes stroke width; and $z_1$ encodes other digit styles.

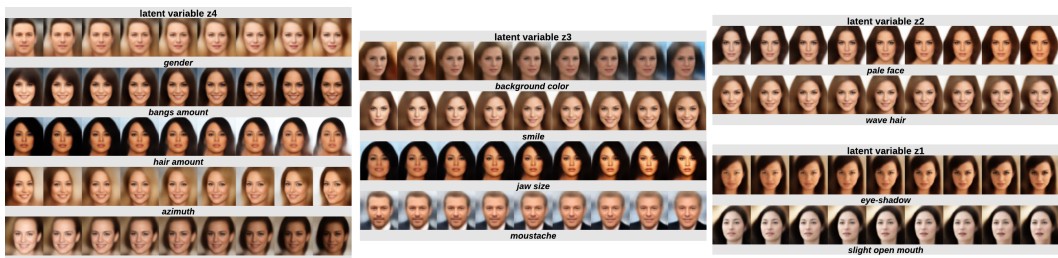

Figure 7: Visualization of hierarchical features learnt for CelebA data. Each subfigure is generated by traversing along a selected latent dimension in each row within each hierarchy of $z_l$'s. **From left to right:** latent variables $z_4$ to $z_1$ progressively learn major (*e.g.*, gender in $z_4$ and smile in $z_3$) to minor representations (*e.g.* wavy-hair in $z_2$ and eye-shadow in $z_1$) in a disentangled manner.

during progression and the larger value of $I_{over}$. This suggests that, since the teacher-student model was motivated for progressively learning nuisance variables, the extent to which its capacity can grow for learning new representations is limited by two fundamental causes: 1) because it increases the dimension of the same latent vectors at the same depth, the growth of the network capacity is limited in comparison to *pro*-VLAE, and 2) the Jacobian supervision further restricts the student model to maintain the same disentangling ability of the teacher model.

**Disentangling hierarchical representations:** We also qualitatively examined *pro*-VLAE on data with both relatively simple (MNIST) and complex (CelebA) factors of variations, all done in unsupervised training. On MNIST (Figure 6), while the deepest latent representations encoded the highest-level features in terms of digit identity, the representations learned at shallower levels encoded changes in writing styles. In Figure 7, we show the latent representation progressively learned in CelebA from the highest to lowest levels of abstractions, along with disentangling within each level demonstrated by traversing one selected dimension at a time. These dimensions are selected as examples associated with clear semantic meanings. As shown, while the deepest latent representation $z_4$ learned to disentangle high-level features such as gender and race, the shallowest representation $z_1$ learned to disentangle low-level features such as eye-shadow. Moreover, the number of distinct representations learned decreased from deep to shallow layers. While demonstrating disentangling by traversing each individual latent dimension or by hierarchically-learned representations has been separately reported in previous works (Higgins et al. (2017); Zhao et al. (2017)), to our knowledge this is the first time the ability of a model to disentangle individual latent factors in a hierarchical manner has been demonstrated. This provides evidence that the presented progressive strategy of learning can improve the disentangling of first the most abstract representations followed by progressively lower levels of abstractions.

## 5 CONCLUSION

In this work, we present a progressive strategy for learning and disentangling hierarchical representations. Starting from a simple VAE, the model first learn the most abstract representation. Next, the model learn independent representations from high- to low-levels of abstraction by progressively growing the capacity of the VAE deep to shallow. Experiments on several benchmark data sets demonstrated the advantages of the presented method. An immediate future work is to include stronger guidance for allocating information across the hierarchy of abstraction levels, either through external multi-scale image supervision or internal information-theoretic regularization strategies.

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

APPENDIX

## A  EXAMPLES OF PERFORMANCE OF DIFFERENT METRICS

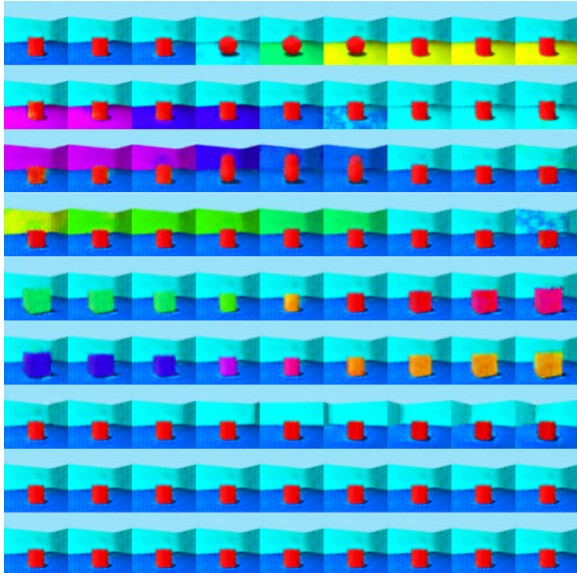

**MIG: 0.1295, MIG-sup:0.4225, factor: 0.6653**

Figure 8: An example of one factor being encoded in multiple dimensions. Each row is a traverse for one dimension (dimension order adjusted for better visualization). Notice that both dim1 and dim2 are encoding floor-color, both dim3 and dim4 are encoding wall-color, and both dim5 and dim6 are encoding object color. Therefore, the MIG is very low since it penalizes splitting one factor to multiple dimensions. On the other hand, the MIG-sup and factor-metric is not too bad since one dimension mainly encodes one factor, even though there are some entanglement of color-vs-shape and color-vs-scale.

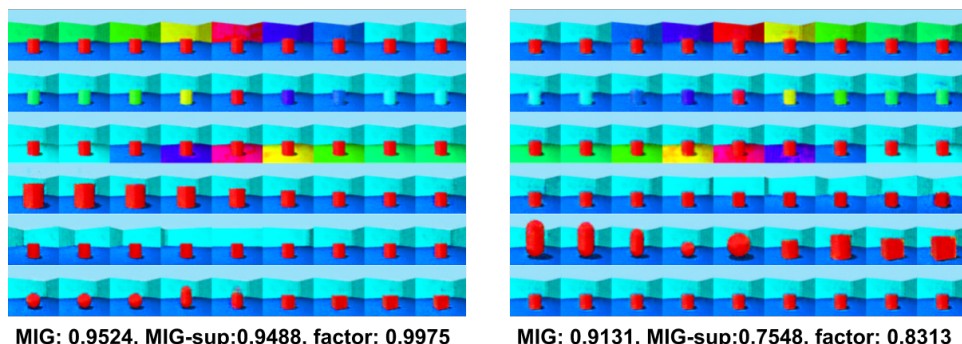

**MIG: 0.9524, MIG-sup:0.9488, factor: 0.9975**     **MIG: 0.9131, MIG-sup:0.7548, factor: 0.8313**

Figure 9: An example of one dimension containing multiple factors. Each row is a traverse for one dimension (dimension order adjusted for better visualization). Notice that both models achieve high and similar MIG because all 6 factors are encoded and no splitting to multiple dimensions. However, the right-hand side model has much lower MIG-sup and factor-metric than the left-hand side model. Because both scale and shape are encoded in dim5, while dim6 has no factor. Both MIG-sup and factor-metric penalize encoding multiple factors in one dimension. Besides, our MIG-sup is lower and drops more than factor-metric because MIG-sup is stricter in this case.

# B A CLOSER COMPARISON WITH VLAE

## B.1 TWO DIMENSIONAL TRAVERSING ON MNIST DATASET

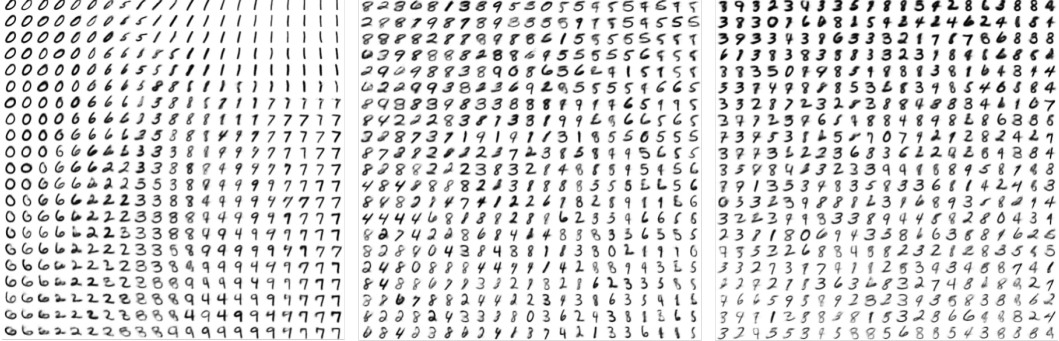

Figure 10: MNIST traversing results following the same generation strategy and network hierarchy as those presented in Figure 5 of (Zhao et al. (2017)). The network has 3 layers and 2 dimensional latent code at each layer. Each image is generated by traversing each of the two-dimensional latent code in one layer, while randomly sampling from the other layers. **From left to right:** The top layer $z_3$ encodes the digit identity and tilt; $z_2$ encodes digit width (digits around top-left are thicker than digits around bottom-right); and the bottom layer $z_1$ encodes stroke width. Compared to VLAE, the representation learnt in the presented method suggests smoother traversing on digits and similar results for digit width and stroke width.

## B.2 INFORMATION ALLOCATION IN EACH LAYER

Table 1: Mutual information $I(x; z_l)$ between data $x$ and latent codes $z_l$ at each $l$-th depth of the network, corresponding to the qualitative results presented in Fig. 4 and Fig. 6 on 3Dshapes and MNIST data sets. Both VLAE and the presented $pro$-VLAE models have the same hierarchical architecture with 3 layers and 3 latent dimensions for each layer. Compared to VLAE, the presented method allocates information in a more clear descending order owing to the progressive learning.

| 3DShapes | $I(x; z_3)$ | $I(x; z_2)$ | $I(x; z_1)$ | total $I(x; z)$ |
|---|---|---|---|---|
| VLAE | 4.41 | 4.69 | 5.01 | 12.75 |
| $pro$-VLAE | 6.94 | 6.07 | 0.00 | 13.02 |

| MNIST | $I(x; z_3)$ | $I(x; z_2)$ | $I(x; z_1)$ | total $I(x; z)$ |
|---|---|---|---|---|
| VLAE | 8.28 | 8.89 | 7.86 | 11.04 |
| $pro$-VLAE | 9.83 | 8.24 | 6.28 | 10.93 |

## C    ABLATION STUDY ON IMPLEMENTATION STRATEGIES

Table 2: The effect of progressive implementation strategies, *i.e.*, "fade-in" and pre-trained KL penalty, on successful training rates. We conducted 15 ablations experiments that each has 4 sub-experiments. As shown, the *pro*-VLAE cannot be trained successfully without the presented implementation strategies, while each of the strategies helps stabilize the progressive training.

| no strategies | pre-trained KL only | fade-in only | Both |
|---|---|---|---|
| 0.0 | 0.667 | 0.733 | 0.867 |

## D    CLOSER INVESTIGATION OF INFORMATION FLOW OVER LATENT VARIABLES

In this section, we present additional quantitative results on how information flow among the latent variables during progressive training. We conducted experiments on both 3DShapes and MNIST data sets, considering different hierarchical architectures including a combination of different number of latent layers $L$ and different number of latent dimensions $z_{dim}$ for each layer. Each experiment was repeated three times with random initializations, from which the mean and the standard deviation of mutual information $I(x; z_l)$ were computed.

As shown in Tables 3-8, for all hierarchical architectures, the information amount in each layer is captured in a clear descending order, which aligns with the motivation of the presented progressive learning strategy. Generally, the information also tends to flow from previous layers to new layers, suggesting a disentanglement of latent factors as new latent layers are added. This is especially obvious for 3DShapes data where the generative factors are better defined.

In addition, models with small latent codes ($z_{dim} = 1$) are not able to learn the same amount of information (total $I(x, z)$) as those with larger latent codes ($z_{dim} = 3$). The variance of information in each layer in the former also appears to be high. We reason that it may be because that the model is trying to squeeze too much information into a small code, resulting in large vibrations during progressive learning. On the other hand, while a model has large latent codes ($L = 4, z_{dim} = 3$), the information flow becomes less clear after the addition of certain layers. Overall, assuming there are $K$ generative factors and there are $D$ dimensions in total available in model, ideally we would like to design the model such that $D = K$. However, since $K$ is unknown in most data, $L$ and $z_{dim}$ become hyperparameters that need to be tuned for different data sets.

Table 3: 3DShapes, $L = 2, z_{dim} = 3$

| progressive step | $I(x; z_2)$ | $I(x; z_1)$ | total $I(x; z)$ |
|---|---|---|---|
| 0 | $10.68 \pm 0.19$ | | $10.68 \pm 0.19$ |
| 1 | $7.22 \pm 0.30$ | $5.94 \pm 0.26$ | $12.88 \pm 0.20$ |

Table 4: 3DShapes, $L = 3, z_{dim} = 2$

| progressive step | $I(x; z_3)$ | $I(x; z_2)$ | $I(x; z_1)$ | total $I(x; z)$ |
|---|---|---|---|---|
| 0 | $10.16 \pm 0.13$ | | | $10.16 \pm 0.13$ |
| 1 | $9.76 \pm 0.05$ | $7.36 \pm 0.10$ | | $13.00 \pm 0.02$ |
| 2 | $6.83 \pm 1.37$ | $6.66 \pm 0.17$ | $5.80 \pm 0.41$ | $13.07 \pm 0.02$ |

Table 5: 3DShapes, $L = 4$, $z_{dim} = 1$

| progressive step | $I(x; \boldsymbol{z}_4)$ | $I(x; \boldsymbol{z}_3)$ | $I(x; \boldsymbol{z}_2)$ | $I(x; \boldsymbol{z}_1)$ | total $I(x; \boldsymbol{z})$ |
|---|---|---|---|---|---|
| 0 | $4.89 \pm 0.03$ | | | | $4.89 \pm 0.03$ |
| 1 | $4.77 \pm 0.04$ | $3.55 \pm 0.04$ | | | $8.14 \pm 0.09$ |
| 2 | $4.66 \pm 0.04$ | $3.75 \pm 0.04$ | $2.70 \pm 0.10$ | | $10.67 \pm 0.09$ |
| 3 | $4.55 \pm 0.11$ | $3.53 \pm 0.35$ | $2.80 \pm 0.19$ | $2.17 \pm 0.14$ | $11.72 \pm 0.03$ |

Table 6: MNIST, $L = 3$, $z_{dim} = 1$

| progressive step | $I(x; \boldsymbol{z}_3)$ | $I(x; \boldsymbol{z}_2)$ | $I(x; \boldsymbol{z}_1)$ | total $I(x; \boldsymbol{z})$ |
|---|---|---|---|---|
| 0 | $5.86 \pm 1.19$ | | | $5.86 \pm 1.19$ |
| 1 | $3.62 \pm 1.04$ | $4.64 \pm 2.83$ | | $7.62 \pm 2.63$ |
| 2 | $3.88 \pm 0.75$ | $4.99 \pm 0.98$ | $2.37 \pm 0.77$ | $8.25 \pm 1.65$ |

Table 7: MNIST, $L = 3$, $z_{dim} = 3$

| progressive step | $I(x; \boldsymbol{z}_3)$ | $I(x; \boldsymbol{z}_2)$ | $I(x; \boldsymbol{z}_1)$ | total $I(x; \boldsymbol{z})$ |
|---|---|---|---|---|
| 0 | $10.08 \pm 0.10$ | | | $10.08 \pm 0.10$ |
| 1 | $9,97 \pm 0.05$ | $8.03 \pm 0.17$ | | $11.01 \pm 0.04$ |
| 2 | $9.91 \pm 0.04$ | $8.09 \pm 0.07$ | $6.27 \pm 0.02$ | $11.02 \pm 0.02$ |

Table 8: MNIST, $L = 4$, $z_{dim} = 3$

| progressive step | $I(x; \boldsymbol{z}_4)$ | $I(x; \boldsymbol{z}_3)$ | $I(x; \boldsymbol{z}_2)$ | $I(x; \boldsymbol{z}_1)$ | total $I(x; \boldsymbol{z})$ |
|---|---|---|---|---|---|
| 0 | $10.06 \pm 0.22$ | | | | $10.06 \pm 0.22$ |
| 1 | $10.11 \pm 0.06$ | $7.95 \pm 0.08$ | | | $10.98 \pm 0.02$ |
| 2 | $10.06 \pm 0.08$ | $8.1 \pm 0.04$ | $6.39 \pm 0.12$ | | $10.98 \pm 0.06$ |
| 3 | $9.99 \pm 0.09$ | $8.11 \pm 0.03$ | $6.45 \pm 0.15$ | $3.52 \pm 0.07$ | $11.03 \pm 0.03$ |

