# OpenReview forum: "PROGRESSIVE LEARNING AND DISENTANGLEMENT OF HIERARCHICAL REPRESENTATIONS"
_ICLR.cc/2020/Conference — Accept (Spotlight)_

### Official Review · AnonReviewer3 · 2019-10-20
**Official Blind Review #3**

**Rating:** 8

**Review:**

The paper proposes an approach to incrementally learn hierarchical representations using a variational autoencoder (VAE). This is shown to be useful qualitatively and quantitatively in terms of disentanglement in the representations.

To learn the hierarchy, the authors use a ladder architecture based on variational ladder autoencoder (VLAE) but incrementally activate the lateral connections across the layers at varying depth of the encoder and the decoder. A vanilla VAE is first trained. Followed by adding stochastic later connections and then retraining the updated architecture. This combined with beta-VAE inspired upweighting of the KL term leads to learning a hierarchy of representations. Each level of the hierarchy, the representations are disentangled.

Inspired by progressive GANs, the authors employ ````"fade-out" when traversing the hierarchy.

The authors also introduce a new metric to capture the one-to-one mapping of the ground truth factors to the latent dimensions.

Ablation studies by varying/removing fadeout compared to incremental learning will be useful. Can fade-out (different weighting of each level) be added directly to VLAE without incremental learning?

Overall the paper is well motivated and easy to read. The results look impressive and the learned hierarchy and latent traversals are convincing. A more thorough comparison with VLAE will make the paper stronger.



**Experience Assessment:**

I have published in this field for several years.

**Review Assessment: Checking Correctness Of Derivations And Theory:**

I assessed the sensibility of the derivations and theory.

**Review Assessment: Checking Correctness Of Experiments:**

I assessed the sensibility of the experiments.

**Review Assessment: Thoroughness In Paper Reading:**

I read the paper thoroughly.

---

> ### Author Response · Authors · 2019-11-12
> **Re: Review #3**
>
> Thank you for your positive and constructive comments.
>
> As suggested, we conducted ablation studies to investigate the performance of our implementation strategies, i.e. “fade-in” and pre-trained KL penalty. The primary purpose of these implementation strategies is to improve the stability of the training, so as to avoid problems such as gradient explosion when adding new layers. Therefore, we focused our study on the effect of these implementation strategies on successful training rates. The results below are summarized from a total of 15 experiments.
>
> No strategies | fade-in only | pre-KL only | both
> 0.0                    |       0.667       |      0.733       |0.867
>
> As shown, both implementation strategies helped improve the training stability of progressive learning. We have added this result and discussion in Appendix C.
>
> As to the varying weights for each ladder layer, we think it is an interesting approach to investigate. We conducted a preliminary experiment on 3DShapes dataset, in which we “fade-in” each layer from 0 to different maximum weights ([10,5,1] for [z3,z2,z1]) in parallel. No obvious improvement has been observed compared to vanilla VLAE in terms of both disentanglement (MIG = 0.41, MIG-sub = 0.55, when beta=10, which is the best beta for VLAE in our experiments) and hierarchical representation.
>
> We also added multiple closer comparisons to VLAE in appendix B, including training and generating a new network on MNIST following the same process as described in Fig 5 of [1] and a detailed quantitative comparison of the mutual information between data and latent codes at different depths. These new results further demonstrate the advantages of the presented methods.
>
> [1] Learning Hierarchical Features from Generative Models, Zhao et al., ICML 2017

---

### Official Review · AnonReviewer2 · 2019-10-23
**Official Blind Review #2**

**Rating:** 8

**Review:**

This paper introduce pro-VLAE, an extension to VAE that promotes disentangled representation learning in a hierarchical fashion.
Encoder and decoder are made of multiple layers and latent variables are not only present in the bottleneck but also between intermediate layers; in such a way, it is possible to encode information at different scales, hence the hierarchical representation. Latent variables can be learned in an incremental way, by making them visible to the whole model progressively, so that as more latent variables become available, they encode lesser and lesser abstract factors.

Experiments are carried out on two benchmarks for disentanglement with annotations and pro-VLAE is compared to other methods in the state of the art.
Here, the authors introduce an extension of the Mutual Information Gap (MIG) metric, namely MIG-sup: it penalizes when multiple generative factors are encoded in the same latent variable. Qualitative results are also shown for 2 non-annotated datasets.

PROS
- The idea is fresh, well explained and experiments are sufficiently thorough. The novelty introduced is enough, provided that not much literature has explored progressive representation learning in the context of disentanglement.
- Results suggest that this is a promising direction for disentangling representations as pointed out by the authors in the conclusions.
- We appreciated the smart solutions for what concerns the implementation and training stabilization.

COMMENTS/IMPROVEMENTS
To improve the quality of the paper, consider the following comments:

- For the sake of completeness, experiments on Information flow should be also quantitative: it would be interesting to see how the information is captured by the latent variables on average on multiple runs, possibly trying different numbers of latent variables z_i.
- In sec 3.1 "z from different abstraction" is too vague and should be better formalized.
- In sec 2: "the presented progressive learning strategy provides an entirely different approach to improve disentangling that is ORTHOGONAL to these existing methods and a possibility to augment them in the future.": you should change to 'different'.

**Experience Assessment:**

I have read many papers in this area.

**Review Assessment: Checking Correctness Of Derivations And Theory:**

I assessed the sensibility of the derivations and theory.

**Review Assessment: Checking Correctness Of Experiments:**

I assessed the sensibility of the experiments.

**Review Assessment: Thoroughness In Paper Reading:**

I read the paper at least twice and used my best judgement in assessing the paper.

---

> ### Author Response · Authors · 2019-11-12
> **Re: Review #2**
>
> Thanks for reviewing our work and the constructive comments.
>
> To further assess the information flow during progressive learning, as suggested, we conducted experiments on hierarchical settings with different combinations of the number of layers L and number of latent dimensions z_dim in each layer. Each experiment was repeated 3 times with random initializations, from which the mean and the standard deviation of mutual information I(x;z_l) were computed. Here we present the results on 3DShapes dataset.
>
> L=2, z_dim=3
> Progressive step |      I(x;z2)     |      I(x;z1)      |      total I(x;z)
>            0                  | 10.68±0.19  |                       |  10.68±0.19
>            1                  |  7.22±0.30   | 5.94±0.26     |  12.88±0.20
>
> L=3, z_dim=2
> Progressive step |      I(x;z3)     |      I(x;z2)      |     I(x;z1)   |    total I(x;z)
>            0                  | 10.16±0.13  |                       |                   |  10.16±0.13
>            1                  |  9.76±0.05   | 7.36±0.10     |                   |  13.00±0.02
>            2                  |  6.83±1.37   | 6.66±0.17     | 5.80±0.41 |  13.07±0.02
>
> L=4, z_dim=1
> Progressive step |      I(x;z4)     |      I(x;z3)      |     I(x;z2)   |    I(x;z1)    |   total I(x;z)
>            0                  |  4.89±0.03   |                       |                   |                    |  4.89±0.03
>            1                  |  4.77±0.04   | 3.55±0.04     |                   |                    |  8.14±0.09
>            2                  |  4.66±0.04   | 3.75±0.04     | 2.70±0.10 |                    | 10.67±0.09
>            3                  |  4.55±0.11   | 3.53±0.35     | 2.80±0.19 | 2.17±0.14  | 11.72±0.03
>
> As shown, for all hierarchical settings, there is a clear descending order of the information amount in each layer, which aligns with the motivation of progressive learning. Besides, the information tends to flow from previous layers to new layers, suggesting a disentanglement of latent factors as new latent dimensions are added to the network, similar to what we presented (Fig 5) in the paper. We have added these results along with additional results on MNIST in Appendix D.
>
> Last but not least, we have modified our paper to better formalize the two statements as suggested.

---

### Official Review · AnonReviewer1 · 2019-10-24
**Official Blind Review #1**

**Rating:** 6

**Review:**

This paper proposed a method for training Variational Ladder Autoencoder (VLAE) using a progressive learning strategy. In comparison to the generative model using a progressive learning strategy, the proposed method focuses not only on the image generation but also on extracting and disentangling hierarchical representation.

Overall, I think the purpose of this paper should be written clearly. It is not clear whether the purpose is learning the disentangled representation or the hierarchical representation. In my opinion, I think the focus of the proposed method lies in the hierarchical representation through progressive learning, but the experiments are involved more with disentanglement. Furthermore, I believe the authors need to explain the relationship between hierarchical representation and disentangled representation. In particular, it is not clear why learning hierarchical representation is helpful for disentangled representations.

The qualitative experiments are not convincing since the proposed model looks worse in both the reconstruction and hierarchical disentanglement for MNIST dataset than the base model VLAE, as shown in Figure 5 in [1]. Regarding the metric used in the experiments, the authors mention that the proposed disentanglement metric MIG-sup is what they first developed for one-to-one property, but it seems that it was already proposed in [2]. In addition, the proposed metric requires ground truth for the generative factors, so its usage is limited and not practical.

I think this work is similar to [3] in that both learn disentangled representations by progressively increasing the capacity of the model. I think the authors need to discuss about this work.

Ablation studies should be presented to verify the individual effects of the progressive learning method and implementation strategies on performance, respectively.

In Figures 2 and 3, the performance gap in the reconstruction error of the proposed method is greater than the base model when beta changes from 20 to 30. Therefore, it is necessary to show if it is robust against the hyperparameter beta.

There is no definition of v_k in Equation (12), so it is difficult to understand the proposed metric clearly.

In summary, I do not think the paper is ready for publication.

[1] Learning Hierarchical Features from Generative Models, Zhao et al., ICML 2017
[2] A Framework for the Quantitative Evaluation of Disentangled Representations, Eastwood et al., ICLR 2018
[3] Understanding disentangling in beta-VAE, Burgess et al., NIPS 2017 Workshop on Learning Disentangled Representations


-------------------------------------
After rebuttal:

Thanks for the revision of the paper and the additional experiments.

The authors' comments and further experiments address most of my concerns. In particular, new experiments show that pro-VLAE performs quantitatively and qualitatively better than VLAE. Also, Figure 10 and the result of the information flow experiment using MNIST show that the first layer learns the intended representations properly.

I appreciate the authors’ efforts put into the rebuttal, and the results of additional experiments are reasonably good. Therefore, I increase my final score to 6: Weak Accept.

**Experience Assessment:**

I have read many papers in this area.

**Review Assessment: Checking Correctness Of Derivations And Theory:**

I carefully checked the derivations and theory.

**Review Assessment: Checking Correctness Of Experiments:**

I carefully checked the experiments.

**Review Assessment: Thoroughness In Paper Reading:**

I read the paper at least twice and used my best judgement in assessing the paper.

---

> ### Comment · AnonReviewer1 · 2019-10-26
> **Additional comments**
>
> I have additional comments on the proposed model.
>
> Based on Figures 4, 5, and 6, it seems that the first layer is not as well trained as VLAE. Therefore, as shown in Figure 5, it is necessary to add an experiment to compare mutual information of each layer with VLAE.
>
> The authors claim that in the experiment with MNIST, the first layer learned the factors related to letter style. However, in Figure 6, it is difficult to determine whether the first layer is successfully trained. For the results of the experiment on MNIST, it would be helpful to measure the amount of mutual information for each layer.

---

> > ### Author Response · Authors · 2019-11-12
> > **Re: Additional comments**
> >
> > Thanks for the additional comments.
> >
> > For the results from 3DShape in Figs 4 and 5, the dataset has in-total 6 generative factors while the VLAE/pro-VLAE being tested have 9 latent dimensions available. In this case, an ideal disentanglement should result in only 6 active latent-dimensions and 3 latent dimensions encoding nothing. This was achieved by the presented method but not VLAE, highlighting the improvement brought by the presented progressive learning method which is also quantitatively verified by the metrics included in Fig 4. In other words, the outcome of 3 “inactive” dimensions in Figs 4-5 actually is a desired outcome and demonstrated the advantage rather than disadvantage of the presented method. To further address the reviewer’s concern, we carried out additional experiments on 3DShapes with the presented pro-VLAE using only six latent codes (see appendix D), both in the form of two layers of three-dimensional latent codes and three layers of two-dimensional latent codes. In either case, the presented model obtained a similar amount of total information and no layer was empty.
> >
> > As suggested, for results in Fig 4 and Fig 6, we added 1) the measure of mutual information for each layer as well as 2) the comparison to VLAE model in the Appendix B.
> >
> > 3D shapes, L=3, each z_i has 3 dimensions
> >                     I(z3;x) | I(z2;x) | I(z1;x) | total I(z;x)
> > VLAE             4.41   | 4.69    |  5.01    | 12.75
> > pro-VLAE     6.94   | 6.07    |  0.00    | 13.02
> >
> > MNIST, L=3, each z_i has 3 dimensions
> >                     I(z3;x) | I(z2;x) | I(z1;x) | total I(z;x)
> > VLAE           8.28    | 8.89    |  7.86   | 11.04
> > pro-VLAE    9.83    | 8.24    | 6.28    | 10.93
> >
> > In 3DShapes, as explained above, the latent codes in the shallowest layer of the pro-VLAE were not “active” — this is desired given that there are only six true factors in the dataset and they have been completely captured in the first two layers of the latent codes. In MNIST, I(z1;x) provides quantitative evidence that, in Fig 6, some style information is indeed encoded in that layer. It further confirms that “inactive” latent codes as observed in Figs 4-5 in 3DShape was not a general result but a specific outcome in the case when the true generative factors have been completely captured in the earlier layer. Overall, compared to VLAE, the presented method achieves a clearer descending order of allocation of information for each layer owing to the properties of progressive learning.
> >
> > The quantitative results of mutual information presented in the main text of the paper (Fig 5) were mainly intended to track the progression of the progressive learning, which do not apply to VLAE. Therefore, we added the above results of mutual-information in section B of the Appendix as a closer comparison with VLAE.

---

> ### Author Response · Authors · 2019-11-07
> **Re: Review1 (part2)**
>
> In terms of the connection with [3], at a philosophical level, we acknowledge that our work and [3] loosely share a similar motivation that the most abstract representations can be learned first before others. However, the two approaches are entirely different, two of the most important differences being 1) the definition of the “capacity” of the VAE and 2) the progressive learning strategy. First, the capacity in [3] was defined by the information capacity of the latent code and controlled by the KL divergence of the latent distribution to an isotropic Gaussian prior (nats). In comparison, the capacity in this paper is defined and controlled by the trainable parameters and growable architectures of the neural network. Second, the capacity in [3] was increased by gradually loosening the constraint on the KL divergence loss. In comparison, we propose a completely different strategy of progressive learning that incrementally increase the “capacity” of the network by growing additional latent variables and new parameters of the model in the ladder connections. This was inspired by recent works in growing of neural network’s architectures (which we extend to growing the latent codes) and is completely different from the approach presented in [3].
>
> We indeed attempted to demonstrate how the considered methods are affected by the hyperparameters beta, hence the results presented in Figs 2 and 3. We acknowledge that the presented method is not suitable for a high value of beta. We reason that, since the presented progressive learning strategies already promote disentangling, a high value of beta may over-promote disentangling at the expense of reconstruction quality. We have revised the paper to add this discussion. We however would like to note that, as shown in Fig 2, the presented method outperforms the baseline models with a clear margin in the majority of the hyperparameters tested.
>
> In the original submission, we designed comparison studies with vanilla VAE, VLAE, the teacher-student model, and the presented method, with the intention for an ablation study that shows the effect of the two individual components: i.e., the hierarchical representations, and the progressive training strategy. We are currently working to include additional ablation studies that investigate the effect of the implementation strategies including the fade-in strategy, which we will include once completed.
>
> Last but not least, many thanks for point out the missing definition of v_k. We have added it in the paper.

---

> ### Author Response · Authors · 2019-11-07
> **Re: Review1 (part1)**
>
> We would like to thank the reviewer for the comments. Below we clarify the key confusions and questions raised.
>
> We would like to clarify that the overall purpose of this paper, motivated by “starting small”, is to progressively learn disentangled representations from high- to low-levels of abstractions. Similar to the hierarchical representations in the VLAE model [1], we define the representations at different levels of abstractions (corresponding to different hierarchy of the network). However, as the key contribution of this paper, we learn these representations in a progressive manner from high- to low-levels of abstraction. Our point, therefore, is not that learning hierarchical representations will help learning disentangled representations. Instead, we argue and demonstrate that the presented progressive learning strategy, incrementally extracting generative factors from high- to low-levels of abstraction, will help learning disentangled representations.
>
> We would like to clarify that MNIST images in Fig 5 in [1] and MNIST images in this paper (Fig 6) are generated with two different processes. In Fig 5 in [1], the images were generated by traversing each dimension of the two-dimensional latent code in one layer along with random sampling from other layers. Therefore, the images generated appeared to change smoothly along the x and y axis. In comparison, MNIST images in this paper (Fig 6) were generated by random sampling in one layer while fixing the latent code in all the other layers. This generation strategy was identical to that used in generating Figure 6 and Figure 7 in [1]. To clear the reviewer’s concern, we have generated new MNIST examples following the same strategy as used in Fig 5 in [1] and add the results to the supplemental material. As shown, compared to Fig 5 [1], the generated images on MNIST appears to be better traversing across the digit type, while similar in generating other variations such as width and stroke.
>
> Indeed, the metric proposed in [2] shared a similar motivation to the metric presented in this paper. However, the approaches to the calculation of these two metrics were entirely different.  The metric in [2] was calculated based on training a regressor function f, which is affected by the choice of regressors and its hyperparameters. The drawbacks of this type of approaches have been discussed in [4]. In comparison, the presented approach of metric calculation, similar to MIG, does not involve additional classifiers and is therefore unbiased for the hyperparameter settings. We have modified our paper to discuss [2] and its relation with the proposed metric.
>
> We were uncertain about the criticism that “the proposed metric requires ground truth for the generative factors, so its usage is limited and not practical.” To our knowledge, all recent metrics [2][4][5][6] proposed for measuring disentanglement require ground truth factors. We believe that the presented metric presents a necessary supplement to MIG to capture what is not measured therein, and we were able to demonstrate that in our experiments in Fig 3.
>
> [4] Disentangling by factorising. Hyunjik et. al. ICML 2018.
> [5] Isolating sources of disentanglement in variational autoencoders. Chen et. al. NeurIPS 2018.
> [6] beta-vae: Learning basic visual concepts with a constrained variational framework. Higgins et. al. ICLR 2017

---

### Decision · Program_Chairs · 2019-12-19

**Decision:**

Accept (Spotlight)

**Comment:**

This paper proposes a novel way to learn hierarchical disentangled latent representations by building on the previously published Variational Ladder AutoEncoder (VLAE) work. The proposed extension involves learning disentangled representations in a progressive manner, from the most abstract to the more detailed. While at first the reviewers expressed some concerns about the paper, in terms of its main focus (whether it was the disentanglement or the hierarchical aspect of the learnt representation), connections to past work, and experimental results, these concerns were fully alleviated during the discussion period. All of the reviewers now agree that this is a valuable contribution to the field and should be accepted to ICLR. Hence, I am happy to recommend this paper for acceptance as an oral.